# Extracellular Vesicles, Circulating Tumor Cells, and Immune Checkpoint Inhibitors: Hints and Promises

**DOI:** 10.3390/cells13040337

**Published:** 2024-02-13

**Authors:** Sara Bandini, Paola Ulivi, Tania Rossi

**Affiliations:** Biosciences Laboratory, IRCCS Istituto Romagnolo per lo Studio dei Tumori (IRST) “Dino Amadori”, 47014 Meldola, Italy; sara.bandini2@irst.emr.it (S.B.); tania.rossi@irst.emr.it (T.R.)

**Keywords:** circulating tumor cells, extracellular vesicles, immunotherapy, immune checkpoint inhibitors

## Abstract

Immune checkpoint inhibitor (ICI) therapy has revolutionized the treatment of cancer, in particular lung cancer, while the introduction of predictive biomarkers from liquid biopsies has emerged as a promising tool to achieve an effective and personalized therapy response. Important progress has also been made in the molecular characterization of extracellular vesicles (EVs) and circulating tumor cells (CTCs), highlighting their tremendous potential in modulating the tumor microenvironment, acting on immunomodulatory pathways, and setting up the pre-metastatic niche. Surface antigens on EVs and CTCs have proved to be particularly useful in the case of the characterization of potential immune escape mechanisms through the expression of immunosuppressive ligands or the transport of cargos that may mitigate the antitumor immune function. On the other hand, novel approaches, to increase the expression of immunostimulatory molecules or cargo contents that can enhance the immune response, offer premium options in combinatorial clinical strategies for precision immunotherapy. In this review, we discuss recent advances in the identification of immune checkpoints using EVs and CTCs, their potential applications as predictive biomarkers for ICI therapy, and their prospective use as innovative clinical tools, considering that CTCs have already been approved by the Food and Drug Administration (FDA) for clinical use, but providing good reasons to intensify the research on both.

## 1. Introduction

The advent of immunotherapy in cancer treatment, which aims at improving natural defenses against malignant cells, has revolutionized oncology research, bringing new hope to cancer patients [1]. While several approaches are encompassed in immunotherapy, including cancer vaccines, cytokine therapies, and oncolytic virus therapies, the administration of immune checkpoint inhibitors (ICIs) is now entering clinical practice as one of the most important immune therapies, alone or in combination with radiotherapy and chemotherapy [2,3]. 

Immune checkpoints exist in a variety of molecules expressed by immune and tumor cells, with a role in acquired immune response inhibition or activation, such as cytotoxic T lymphocyte-associated molecule 4 (CTLA-4), programmed cell death ligand 1 (PD-L1), and programmed cell death receptor 1 (PD-1) [4]. In normal conditions, immune checkpoints engage with other partner proteins to inhibit T cell functions, while blocking their binding through ICI administration ideally results in tumor control and clearance by improving the tumor cells’ immunogenicity and sensitivity to cell killing (2). Different cancer types, including melanoma, non-small-cell lung cancer, renal cell carcinoma, and breast cancer [5,6,7,8,9], have demonstrated sustained clinical response rates. However, only a small fraction of cancer patients actually benefits from ICI therapy, and the underlying mechanisms are still far from being fully elucidated [10]. Thus, the need for reliable biomarkers that are able to enhance patient stratification has become imperative.

In recent years, the characterization of analytes retrievable in the liquid compartments of the human body, the so-called “liquid biopsy”, has emerged as a promising, minimally invasive tool for screening, early diagnosis, minimal residual disease (MRD) assessment, therapy resistance monitoring, and also as a biomarker surrogate [2]. In this context, peripheral blood-derived extracellular vesicles (EVs) and circulating tumor cells (CTCs) are among the most researched due to their pivotal role in both the tumor-resident and circulating microenvironment, and pre-metastatic niche modulation [11]. Moreover, EVs and CTCs have the ability to modulate the immune response, by directly and indirectly communicating with blood cells, among which are those ascribable to the immune system [12,13].

In this review, we will outline the involvement of EVs and CTCs in the immune response and tumor microenvironment (TME) modulation, focusing on their communication with immune cells, which represent the cellular underpinnings of immunotherapy. From a translational perspective, we will report the most recent findings concerning their role as predictive biomarker surrogates in different cancer types, as well as future applications. 

## 2. Immune Checkpoint Inhibitors

The most important role of the immune system relies on the recognition of “foreign” cells from self-cells. In this context, “checkpoints” are exploited by the immune system to selectively target foreign cells. Immune cells that express immune checkpoints must be activated or inhibited to start an immune response. ICIs are drugs that work in this way [14]. They are usually formulated as antibodies, whose working principle relies on blocking the checkpoint-related proteins expressed by T cells and by some types of cancer cells. More specifically, they do not exert a growth-inhibitory effect against tumor cells, but they empower the previously established immune response [15]. Over the past few years, lots of inhibitory immunoreceptors have been discovered and investigated, such as PD-1, CTLA-4, LAG3, TIM3, TIGIT, and BTLA. Antibodies targeting immune inhibitory receptors, such as CTLA-4, PD-1, and PD-L1, have been the most widely used immunotherapeutic agents in the last decade [16].

### 2.1. PD-1 Inhibitors

The PD-1 checkpoint is expressed on the surface of several activated immune cells, including macrophages, dendritic cells (DCs), Langerhans cells, B cells, and T cells, and plays a fundamental role in the regulation of T cell-mediated responses through programmed death signaling [17]. Antibodies targeting the PD-1 pathway have revolutionized the treatment management of different cancers, such as Merkel cell carcinoma (MCC), melanoma, head and neck squamous cell carcinoma (HNSCC), and non-small-cell lung cancer [18].

PD-1 is a receptor localized on the surface of T cells, and its natural ligands are PD-L1 and PD-L2 expressed by tumor cells. PD-1 and its ligands have been shown to play a key role in helping tumors resist immunity-induced apoptosis, resulting in tumor progression, and their binding activates downstream signaling pathways leading to the inhibition of T cell activation [19,20]. More specifically, after the presentation of tumor antigens by the major histocompatibility complex (MHC) class I, CD8+ T cells release interferon (IFN)-γ, which binds to its receptor on cancer cells. This event elicits the expression of PD-L1 via the IRF1 transcription factor in cancer cells, which in turn binds to PD-1 on the T cell surface, finally culminating in T cell inhibition. Anti-PD-1 antibodies, such as Nivolumab and Pembrolizumab, selectively target the interaction between PD-1 receptors on CD8+ T cells and its ligands PD-L1/PD-L2 expressed by cancer cells, resulting in the abolishment of the inhibition of the CD8+ T cells and the restoration of antitumor activity [19,20]. 

The fully human Immunoglobulin G4 (igG4) monoclonal antibody (mAb) Nivolumab (BMS-936558, ONO-4538, or MDX1106, trade name Opdivo; Bristol-Myers Squibb, Princeton, NJ, USA) has been approved by the Food and Drug Administration (FDA) for the treatment of different tumor types, such as melanoma, renal cancer, and squamous and non-squamous non-small-cell lung carcinoma (NSCLC) [21]. 

Another IgG4 mAb that acts by disrupting the PD-1/PD-L1 axis is Pembrolizumab (Keytruda, Merck, Rahway, NJ, USA), which has been approved by the FDA for the treatment of several malignancies [22]. Moreover, Pembrolizumab was recently approved by the FDA as the first tissue-agnostic/site-agnostic drug for the treatment of patients with mismatch repair deficient/metastatic microsatellite instability—high (dMMR/MSIH) [23]. 

Lastly, the fully humanized IgG4 mAb Cemiplimab (Libtayo^®^, Regeneron Pharmaceuticals Inc. Westchester County, NY, USA; Sanofi, Paris, France) blocks the interaction of PD-1 with PD-L1/PD-L2 and has been approved by the FDA for the treatment of patients diagnosed with metastatic or locally advanced cutaneous squamous cell carcinoma who are considered ineligible for curative surgery or radiotherapy [24].

### 2.2. PD-L1 Inhibitors

PD-L1 and PD-L2 are the two ligands for the PD-1 receptor [25]. As reported above, their binding results in tumor progression due to T cell inhibition. It has been shown that PD-L1 can be expressed on the surface of both tumor and immune cells, and the detection of its expression is a powerful biomarker to predict the response to anti-PD-1/PD-L1 therapies in patients with different malignancies [26]. 

Currently, three PD-L1 inhibitors, Atezolimumab, Durvalumab, and Avelumab, have been approved by the FDA for the treatment of some solid tumors, including NSCLC, HNSCC, melanoma, and MCC.

The mechanism of action of these antibodies relies on blocking the interaction of PD-L1 with PD-1 and CD80. However, the fragment crystallizable (Fc) region of Atezolizumab and Durvalumab is modified in order to eliminate antibody-dependent cellular cytotoxicity, thus preventing the depletion of T cells expressing PD-L1 [27]. Instead, Avelumab holds the native Fc region, which can engage the Fc-γ receptors expressed on natural killer (NK) cells, inducing antibody-dependent cellular cytotoxicity [27]. The FDA approved Atezolizumab for the treatment of patients diagnosed with localized and metastatic urothelial carcinoma due to the results obtained from a phase II clinical trial, which showed overall response rates of 10% in patients whose disease had progressed after platinum-based chemotherapy treatment [28]. On the basis of these results, a series of studies have been carried out to evaluate the antitumor efficacy of anti-PD-L1 antibodies in combination with different chemotherapeutic regimens. For the treatment of other malignancies, currently, the FDA has approved the use of Atezolizumab in combination with chemotherapy for the treatment of squamous and non-squamous NSCLC, small-cell lung cancer, and PD-L1-positive triple-negative breast cancer [7,29,30,31]. 

For patients diagnosed with urothelial carcinoma, the FDA has also approved the use of Durvalumab and Avelumab [32,33,34]. Moreover, Avelumab can also be administered to patients with Merkel cell carcinoma [35]. Several clinical trials are ongoing to evaluate the efficacy of anti-PD-L1 antibodies in combination with other immunotherapies and chemotherapies.

### 2.3. CTLA-4 Inhibitors

CTLA-4 acts as a co-inhibitory receptor. It is expressed primarily by T cells, with constitutive expression on regulatory T cells (Tregs). After the recognition of specific antigens, intracellular CTLA-4 is translocated to the cell surface, where it competes with CD28 to interact with B7 molecules on antigen-presenting cells (APCs), exhibiting a higher affinity. This interaction induces negative signals to T cells, and drives to attenuation of T cell proliferation, activation, and overall function [36,37]. 

In syngeneic mouse models, it has been shown that treatment with anti-CTLA-4 antibodies can induce a significant and long-term regression of established tumors [38].

The human IgG1 mAb Ipilimumab (Yervoy) was developed to block the function of CTLA-4. In 2011, it was first approved for the treatment of melanoma [39], and later for the treatment of advanced renal cell carcinoma, MSI-H/dMMR metastatic colorectal cancer, malignant pleural mesothelioma, NSCLC, and hepatocarcinoma, in combination with Opdivo (Nivolumab) [40]. In 2020, the FDA declared that patients with unresectable malignant pleural mesothelioma (MPM), and NSCLC (with tumor PD-L1 expression ≥ 1% and without EGFR/ALK alterations) can benefit from treatment with Opdivo (Nivolumab) plus Yervoy (Ipilimumab) as a first-line treatment [41].

The development of anti-CTLA-4 ICIs is challenging because it has been demonstrated that monotherapy is less effective with respect to PD-1/PD-L1 inhibitors, with higher rates of serious immune-related adverse events. These differences may arise from the specific roles played by CTLA-4 and PD-1/PD-L1 checkpoints in the immune system, such as the response to cancer cells [42]. 

In the last few years, a growing body of research has identified a number of novel immune checkpoint targets, such as NKG2A ligands, TIGIT, B7-H6 ligands, galectin 3, TIM3, and others. Several studies are ongoing to clarify their potential role in the clinic [43].

## 3. EVs 

### 3.1. Biology and Genesis of Extracellular Vesicles

The term extracellular vesicle is used globally to identify a family of nanoparticles made up of a lipid bilayer, which are heterogeneous in regard to their content, surface composition, and size, and are released by most cells [44]. The definition, therefore, includes all vesicles with a diameter between 40 nm and 5 µm, which are divided mainly related to their size or different modes of biosynthesis into apoptotic bodies, microvesicles and exosomes [45]. Initially, it was thought that the only function of these vesicles was to collect and direct the waste materials no longer desired by the cells towards the lysosomes; then, only in 2006, with the discovery of their contents, did they acquire interest as fundamental mediators of cell–cell communication, which is involved in both physiological and in harsher conditions, such as those representing the tumor microenvironment [46]. Currently, the analysis of EVs relies on the guidelines provided by the International Society for Extracellular Vesicles (ISEV). These guidelines, released in 2014 and updated in 2018, provide a series of standards that should be adopted by researchers to support their findings concerning EVs functional analysis and cargo profiling [47,48].

The two major groups of EVs that are most studied are exosomes and microvesicles [49]. Extracellular vesicles are recognized as an important short- and long-distance communication system, since they are able to transport specific sets of biomolecules, whose interaction with target cells can lead to variations in the latter’s activity, migration, proliferation, and survival [50]. The modulation of the physiology of the target cell by EVs can also be achieved thanks to the transfer of membrane proteins or lipids from the vesicle to the cell as a result of the fusion of the vesicle itself with the plasma membrane of the target cell. Alternatively, the release of the vesicle cargo into the cytosol of the target cell can occur following the uptake of the vesicle via endocytosis. That mechanism, in particular, appears to be the most frequent one in the interaction between vesicles and target cells; although the release of the contents of the vesicles through this route can lead to their degradation via the endosomal pathway [51].

More recently it has been demonstrated that EVs can convey not only proteins or lipids, but also microRNAs (miRNAs), pre-miRNAs, other non-coding RNAs, and mRNA [52], which are then translated in the target cell with consequent activation or inhibition of certain cellular processes, depending on the information transferred [53]. It is known that the miRNA-200 family can influence cancer biology in the context of pancreatic adenocarcinoma [54].

Nowadays, extracellular vesicles, along with CTCs and circulating tumor DNA (ctDNA), have become the focus of liquid biopsies, as biomarkers that can yield new viewpoints in the area of cancer diagnosis, prognosis, and treatment [55]. Among all the markers that can be found within a blood sample, the most used biological fluid, exosomes are the most abundant class of extracellular vesicles and have so far aroused the greatest interest in the biomedical field [56]. This category seems to show greater advantages in terms of composition and stability, thanks to which they have gained a strong predictive value. Indeed, the probability of isolating exosomes from the biological fluid under examination is much higher than that of obtaining circulating tumor cells or circulating tumor DNA, since their quantity in circulation is disproportionately higher. Furthermore, exosomes are more stable thanks to their organization in a lipid bilayer. This biological stability is also reflected in long-term conservation [57]. Another important aspect concerns their contents, as they can contain different types of molecules that recall the parental cells from which they originated. In this way they appear to be much more representative of ctDNA, which, instead, provides the information belonging to apoptotic cells [58]. Exosomes were first identified in the collected medium from reticulocyte cell cultures as microvesicles containing membrane proteins, including the transferrin receptor [59]. Subsequently, a plethora of cell types capable of releasing exosomes into the extracellular environment have been described, namely hematopoietic cells (B cells, T cells, dendritic cells, mast cells, and platelets), intestinal epithelial cells, Schwann cells, neuronal cells, adipocytes, and fibroblasts (NIH3T3), as well as tumor cells, have all been observed as cells secreting internal vesicles [60]. Exosomes have also been described by Trams and colleagues, who showed how cultures of various normal and cancerous cell lines produced vesicles with a 5′-nucleotidase activity, which reflected the ectoenzymatic activity of the parent monolayer culture. In vesicles with an enlarged size, a second population of vesicles of about 40 nm in diameter were found and were called exosomes. The work of Trams and colleagues was then followed by that of Harding and Stahl, who described the release of small vesicles and tubules from rat reticulocytes, and a microscopic study describing the exocytosis of bodies of approximately 50 nm [61].

Microvesicle biogenesis is a much less explored mechanism than that paving the way to the formation of exosomes. This type of vesicle appears to be formed by outward budding and fission of the plasma membrane. A combination of factors leads to the formation of MVs, such as the redistribution of phospholipids, including the relocation of phosphatidylserine on the outer leaflet and the contraction of actin-myosin machinery.

Microvesicles have different release mechanisms, which depend on their content. Some of these mechanisms are also common to a class of retroviruses [62], which is why some EVs released by tumor cells are retrovirus-like particles.

Conversely, exosomes derive from the intracellular endosomal compartment. The activation of cell-specific receptors and signaling pathways that initiate exosome synthesis are tightly regulated. 

Exosomal vesicles are initially formed through a process that involves budding towards the inside of the limiting membrane of the early endosome (EE), formed by the fusion of the vesicles’ primary endocytic cells, with the progressive accumulation of the intraluminal vesicles (ILVs) inside them [63]. Secondly, these multivesicular endosomes (known as multivesicular bodies, MVBs) were shown to fuse with the membrane of the cell to release intraluminal vesicles as exosomes, with a size ranging ~40–160 nm in diameter, into the extracellular space [64]. The fate of MVBs is not unique, as they can be targeted both by the lysosomal compartment for degradation and by the plasma membrane as just described. The formation of exosomes requires the coordinated work of various proteins, for example the Rab GTPase proteins that control endosomal trafficking. These proteins might be capable of triggering the release of exosomes, in particular Rab GTPases 27a and 27b. The biogenesis of these vesicles is regulated by the molecular complex known as endosomal sorting complexes required for transport (ESCRT) [65]. This machinery is made up of four multiprotein components, ESCRT-0, ESCRT-I, ESCRT-II, and ESCRT-III. At the beginning of the ESCRT-dependent training process, there is a crossing for cargo delivery, which is determined by the twelfth checkpoint protein ubiquitin (ub). ESCRT-0 plays a role in the recognition of mono-ubiquitinated proteins via an HRS and STAM 1/2 heterodimer [66]. Subsequently, ESCRT-I and ESCRT-II combine with ESCRT-0 to create a recognition domain with a strong affinity for ubiquitinated substrates via the endosomal membrane, where it will eventually bud. Finally, ESCRT-III converges with the complex to detach the membrane and release the buds into the endosome. The de-ubiquitination of the cargo through the de-ubiquitylating enzymes prevents ILVs from being delivered to the lysosome for degradation. It is thought that, in addition to Rab GTPase proteins, the release of exosomes can also be enhanced by hypoxia, which is frequent in tumors. Exosome ESCRT-independent biogenesis is due to the presence of sphingolipid ceramide, which plays a fundamental role in ILV formation; this may allow the generation of raft-based microdomains, which induce a natural negative curvature on the membranes [67]. The MVBs, thus, formed can be directed to the plasma membrane, where fusion with it determines the release of exosomes into the extracellular space. Once it arrives near the target cell, the binding with it is made possible, above all, by the tetraspanin–integrin complex. Furthermore, a proinflammatory environment can increase the expression of receptor molecules, such as ICAM-1, on the membrane surface and this promotes the adhesion between the exosomes and the target cells. Exosomes can release their contents into the cytoplasm of target cells as a consequence of their fusion with the membrane of the target cells. The entry of exosomes can also occur through phagocytosis dependent on actin–cytoskeleton interactions and phosphatidylinositol 3-kinase [68].

### 3.2. EVs’ Role in Metastatic Niche and TME

Since metastasis remains the principal cause of death among cancer patients [69], it is crucial to explore all the mechanisms that lead to the formation of the pre-metastatic niche and, then, metastasis itself. For the effective growth and systemic spread of the tumor, continuous crosstalk is necessary, i.e., dense communication between the tumor cells and the local or distant host environment. 

As stated before, even if tumorigenesis is a cell-autonomous process, it has been proven that extracellular vesicles may influence it by improving cell–cell communication, not only between cancer cells themselves but also between tumor cells and other types of cells, such as immune cells and, finally, between a cancer cell and the microenvironment around which the tumor develops [70,71]. Signaling through exosomes is probably the most renowned way of EV signaling in the context of cancer. In detail, they seem to have an essential role in angiogenesis, cancer progression, TME, and metastasis [72]. Exosomes manage to take part in the process of metastasis development by preparing the environment suitable for the engraftment and the colonization of circulating tumor cells, which, after leaving the primary site, move toward distant secondary organs. This scenario is referred to as the pre-metastatic niche, which has been observed in many kinds of tumors and it is made up of different types of cells, including tumor-derived secreted factors (TDSFs), bone marrow-derived cells (BMDCs), suppressive immune cells, host stromal cells, and EVs [73]. A sweeping force for tumor progression and metastasis development is provided by a chronic inflammatory microenvironment. Indeed, EVs can also affect the tumor microenvironment, which is mainly composed of inflammatory cells, stromal cells, and an extracellular matrix, etc. 

Hoshino reported that exosomes elicit the upregulation of inflammatory molecules in the pre-metastatic niche. In particular, exosome integrins were found to positively regulate the expression of proinflammatory S100 molecules in the distant tissue microenvironment [74]. TDSFs, such as the vascular endothelial growth factor (VEGF), tumor necrosis factor alpha (TNF-α), transforming growth factor-β (TGF-β), and interleukins (ILs), such as IL-6 and IL-10, can derive from tumor cells, which are themselves induced by the local inflammatory microenvironment. These TDSFs in turn affect myeloid cells through paracrine means, in order to stimulate their migration to a future pre-metastatic niche [75]. 

Following stimulation by TDSFs, host stromal cells in the pre-metastatic niche could upregulate the inflammatory factors’ expression. 

As already mentioned, the cells in the immune system, called BMDCs, also arrive in the pre-metastatic niche; it seems that their presence encourages and speeds up the release of inflammatory factors. In addition, exosomes of tumor origin have been shown to bring inflammatory factors into the pre-metastatic niche through the bloodstream. As a result, these mechanisms lead to the generation of an inflammatory microenvironment favorable to tumor cells [76]. 

It also has been demonstrated that extracellular vesicles released by carcinoma-associated fibroblasts (CAFs) may produce a pre-metastatic niche in the lung through the activation of fibroblasts [77].

Several studies show that exosomes are involved in angiogenesis and increase vascular permeability to facilitate the formation of the pre-metastatic niche. For example, exosomes secreted by colorectal cancer (CRC) cells are enriched in miR-25-3p, which promotes the angiogenetic process and targets KLF2 and KLF4, resulting in the disruption of the tight junctions of vascular endothelial cells [78].

To summarize, the mechanisms by which exosomes take part in the formation of the pre-metastatic niche and by which they influence the cells that are part of the tumor microenvironment are multiple. Indeed, they have a bimodal role in cancer: they are able to program the immune system to provoke an antitumor response, but they can also manipulate the local and systemic environment to ensure the growth and dissemination of the tumor. 

### 3.3. EVs’ Role in Immunomodulation

As we all know, the primary function of the immune system is to defend the subject’s body against any external threats. The immune system differentiates the self from the non-self, in particular it tries to attack and, consequently, destroy everything that does not belong to the organism itself and which, for this reason, is potentially dangerous and defined as non-self. In addition to pathogens, the immune system also fights cells in the body that present abnormalities, such as cancer cells.

In the case of cancer, the immune system normally recognizes the antigens found on the surface of tumor cells as non-self and, therefore, begins the process to destroy them. In most cases, unfortunately, the tumor is able to evade the immune response, a phenomenon called ‘immune escape’, by its host using a variety of mechanisms [69].

One mechanism by which tumor cells escape the control of immune system cells is provided by the exosomes released by the tumor itself.

Given that exosomes can interfere with immune responses [68,79], it is also relevant to investigate how they act as mediators between cancer and the immune system.

Here, we aim to highlight the different mechanisms by which exosomes released by tumor cells are able to interfere with the immune response, sometimes inhibiting the cells that take part in it, while others, instead, encourage the release of polypeptide mediators and the generation of specific cells (Figure 1).

#### 3.3.1. CD8+ T Cells

It is renowned that cytotoxic CD8+ T and CD4+ Th1 cells are the principal antitumor immune response effector cells and some studies have reported that cancer-derived exosomes regulate the function of T cells, basically, by damaging proliferation and facilitating the apoptosis of CD8+ T cells.

In particular, scientists have shown that exosomes PD-L1+, highly expressed in tumor tissues, tumor-associated APCs, and stromal cells [80,81,82,83], disseminate directly from the cancer tissue to all over the body, cracking down on the immune system of the patient.

In animal experiments, exosomes carrying PD-L1 promote tumor growth and inhibit the proliferation of CD8+ T cells [84].

Many types of exosomes derived by cancer cells may present the Fas ligand (FasL), whose activation is not only known from the literature to silence the immune response by inducing apoptosis of activated CD8+ T cells, but is also correlated with poor prognosis [85].

The CD8+ T cells’ activity can be compromised by some exosomes carrying miR-498 and miR-3187-3p [86].

#### 3.3.2. Natural Killer (NKs)

Another class of immune cells that act at the level of the antitumor immune response and at the level of immune surveillance are NKs [87]. In order to implement immunological escape, the NK receptor NKG2D must be lost because it usually leads to NK activation. In fact, some studies have highlighted that exosomes that express the NKG2D ligand break down the expression of NKG2D and, thus, inhibit the cytotoxic activity of these cells [88]. 

This kind of NK activity can also be suppressed by the cytokine TGF-β1 present in tumor-derived exosomes [89].

#### 3.3.3. Macrophages

Exosomes can also interfere with the maturation process of some cells belonging to the immune system. Specifically, it has been demonstrated that this class of nanovesicles is able to arrest the maturation of monocytes into macrophages and DCs [90], as reported first by Valenti’s study carried out on exosomes isolated from colorectal and melanoma cancer cells [91]. The way in which exosomes can suppress the immune system is closely related to the presence of some protein components, such as TGF-β, IL-6, and PGE2 [92]. It has been shown that tumor-derived exosomes bring about the inhibition of the differentiation by secreting IL-6, and activating Stata3 signaling too [93].

At the same time, exosomes can inhibit macrophage maturation [94].

It is well known that exosomes derived from glioblastoma stem cells can encourage the monocyte’s differentiation into M2 macrophage, thus resulting in a suppressed immune response. Macrophages, in fact, can be classified according to two subtypes: M1 macrophages, associated with the antitumor response, and M2 macrophages, often involved in pro-tumoral processes and in promoting the growth of cancerous cells [95].

To do this, exosomes exploit the presence of the non-coding RNA within them. For instance, it has been observed that two miRNAs are most relevant in ovarian cancer: miR-222-3p and miR-200b. The level of the first miRNA is higher in ovarian cancer patients and to promote tumor growth it can be transferred into macrophages, in order to induce the tumor promoting M2 population [96]. The second miRNA, miR-200b, is deeply upregulated in the serums from ovarian cancer patients, where it stimulates the proliferation and invasion of tumor cells by promoting the polarization of M2 macrophage [97].

Colon TP53 mutant cancer cells release a huge amount of miR-1246-enriched [98] exosomes. Neighboring macrophages, through the uptake of these exosomes, undergo miR-1246-dependent reprogramming into tumor-associated macrophages (TAMs). Mutp53-reprogammed TAMs foster anti-inflammatory immunosuppression, with increased TGF-β activity [99].

So far, some mechanisms have been reported according to which, through the inhibition of the maturation or activity of various immune cells, exosomes derived from tumor cells are able to repress the activity of the immune system. On the other hand, exosomes are also able to carry out this function by encouraging the generation and proliferation of specific kinds of cells that act in their favor. Furthermore, the local microenvironment means that tumor cells can produce chemokines and cytokines, which, working in synergy with the same exosomes, recruit into secondary organ cells, such as TAMs, tumor-associated neutrophils (TANs), Treg cells, and myeloid-derived suppressor cells (MDSCs) [73], which are capable of inhibiting the antitumor immune response that is normally triggered to try to eradicate cancer [92].

#### 3.3.4. Treg Cells

Exosomes isolated from tumor cells were found to facilitate the generation and expansion of Tregs [100], allowing the immune escape of tumor cells through the release of immunosuppressive cytokines, such as IL-10 and TGF-β1 [101].

Several studies have indicated that exosomes can boost the suppressive function of MDSCs on T cells. For example, in renal cancer, exosomal heat shock protein 70 (HSP70) eases proliferation and enhances the activation of MDSCs via activating TLR2 signaling [102].

### 3.4. Cargo of Extracellular Vesicles

Nowadays, due to their ability to boost antigen-specific immune responses, researchers are focused on the use of exosomes in immunotherapy. Indeed, this occurs because they can transfer an antigen from an APC, such as DCs and tumor cells, to other APCs. All this is also made possible by the wide range of proteins, lipids, and nucleic acids that are part of the exosomes themselves (Table 1).

In particular, the expression profile of exosomal proteins, different in all the various tissues and in the different tumor stages, are strongly associated with cancer development and progression [112]. The increased specificity of exosome cargo can be the basis for determining the cells from which these extracellular vesicles derived. Among all the cargoes, the mRNA content certainly stands out; this very stable type of RNA can carry genetic information on tumor cells; therefore, their presence can be used from a diagnostic and evaluation point of view related to tumor progression and to monitor how the patient responds to the therapy administered [113].

Some of the proteins that are encapsulated within the lumen, or incorporated on the exosome surface, have been defined as possible biomarkers, such as tetraspanin CD63, Alix, and TSG101, while other proteins can be used to distinguish tumor-derived exosomes from non-tumor-derived ones, such as EGFR, EphA2, and EpCAM [112]. Increasing evidence shows that circulating EVs may counter antitumor immunity systemically, since checkpoint ligands, such as PD-L1, CTLA4, and NKG2D, are expressed on their surface.

For example, PD-L1 drives immune checkpoint responses by binding PD-1 on T cells [84]. Another study demonstrated that PD-L1-positive exosomes in blood samples from patients with pancreatic ductal adenocarcinoma can correlate with the worse survival rates (7.8 vs. 17.2 months, *p* = 0.043) [105]. PD-L1 has also been found in blood samples derived from melanoma patients; in this case, exosomal PD-L1 contributes to immunosuppression through CD8+ T cell suppression and is associated with the anti-PD-1 response [84].

Theodoraki et al., using exosomes isolated from head and neck patients’ plasma, found that the levels of PD-L1 carried by exosomes were related to the disease activity and to the clinical stages. Patients with a high frequency of PD-L1+ exosomes in their plasma had a more active and malignant disease than those with low levels [103].

It was also found that serum exosomal PD-L1 levels were effective for predicting anti-PD-1 therapies for patients with NSCLC; furthermore, these levels tend to be associated with survival [104].

Interestingly, exosomal lipids can also be correlated with immunotherapy in cancer. For example, exosomes from B cell lymphoma have been shown to contain the phosphatidylcholine transporter ATP-binding cassette transporter A3 (ABCA3). This kind of exosome may be implicated in resistance to immunotherapy by protecting target cells from treatment with Rituximab, an antibody that attacks the B cell lymphocyte antigen CD20 [106]. 

As already mentioned, the content of exosomes has been proven to be a reliable marker for the selection of cancer patients, in particular plasma-derived exosomal miRNAs.

Some studies confirm the strong importance that miRNAs have reached as biomarkers for the selection of patients with advanced NSCLC. In particular, three miRNAs from the hsa-miR-320 family have been found as potential predictors and hsa-miR-125b-5p has been found to be a potential target for anti-PD-1 treatment, given that it is downregulated in patients with a response to this kind of therapy. The results achieved in this study suggest that patients with low levels of hsa-miR-320d, hsa-miR-320c, hsa-miR-320b, and hsa-miR-125b-5p might better candidates for anti-PD-1 treatment. The continuous decrease in the T cell-suppressor hsa-miR-125b-5p level can be considered as the mark of a better outcome and longer progression-free survival (PFS), due to the T cell function increase [107]. From the data collected, it can be assumed that hsa-miR-320d, hsa-miR-320c, and hsa-miR-320b are to be considered as possible biomarkers, useful for predicting the efficacy of immunotherapy in advanced NSCLCs.

Plasma hsa-miR-200c and hsa-miR-34a levels were also associated with the response and outcome in advanced NSCLC patients treated with anti-PD1 immunotherapy [108].

In addition to miRNAs, exosomes also contain circular RNAs (circRNAs). The latter underlies several mechanisms, through which they control the resistance to some cancer therapies, including immunotherapy. It has been proven that exosome-derived circCCAR1 enhances CD8 + T cell dysfunction and anti-PD1 resistance in patients diagnosed with hepatocellular carcinoma [109]; but also, cancer cell-derived exosomal circUSP7 might induce CD8+ T cell dysfunction and anti-PD1 resistance through the regulation of the miR-934/SHP2 axis in NSCLC patients [111].

The cancer cell-derived circular RNA, circUHRF1, promotes the exhaustion of natural killer cells and might cause resistance to anti-PD1 therapy in hepatocellular carcinoma patients [110]. 

## 4. CTCs 

### 4.1. Biology and Role in the Metastatic Cascade

The metastatic cascade is a multi-step process, through which tumor cells that slough off the primary tumor travel through the bloodstream and reach other distant organs to develop metastases, which represent the major cause of death in oncologic patients [114,115].

In this context, CTCs, a population of rare cells detectable in the peripheral blood of cancer patients, assume a paramount role in guiding the metastatic spread of the solid tumor [116]. Although, currently, the precise mechanisms related to their biology and clinical value still need to be fully elucidated, the awareness of CTC existence and their potential role in the metastatic spread emerged in the XIX century. In fact, the first detection of CTCs dates back to 1869, during an autopsy examination on a metastatic cancer patient. In that year, the Australian physician Ashworth described the presence of cells in the bloodstream resembling the primary tumor, and he assumed that those cells must have passed through the circulatory system to arrive at the vein from which the blood was collected, therefore introducing the concept of CTC [117]. 

The first critical step in the metastatic spread is represented by cancer cell invasion, followed by their entry into the bloodstream, which is commonly known as intravasation. At the first instance, cancer cell invasion can occur passively, with cancer cells sloughed off the edge of the tumor and swept away by the circulation, while active migration can occur, as well as with cancer cells crawling into the vessels, for example due to nutrient and chemokine gradients [118,119]. Active cancer cell invasion was also demonstrated to be triggered by hypoxia [120]. Moreover, intravasation can occur through the following different routes: by entering the bloodstream directly (the hematogenous route), or indirectly, through the lymphatic circulation [121]. Therefore, this process encompasses a number of mechanisms, and several factors orchestrate this phenomenon, dramatically affecting the metastatic cascade [122,123]. In fact, it has been estimated that approximately 1 × 10^6^ cancer cells per gram of tumor tissue are shed in the bloodstream, but the estimated metastasis efficiency is very poor (0.01%) [124]. The multitude of threats to which CTCs are subjected to in the bloodstream are among the main consequences of low metastasis efficiency rates: anoikis, high shear stress causing deformation, fragmentation, cell death, and immune surveillance. As a consequence, CTCs’ half-life is usually very poor: while single CTCs circulate in the bloodstream for approximately 25–30 min, it is estimated that the half-life of CTC clusters is dramatically reduced to 6–10 min [125]. However, despite a lower time span in circulation and a lower paucity compared with single CTCs, it has been shown that CTC clusters have a 23- to 50-fold increased metastatic potential in breast cancer patients [126]. 

Extravasation relies on the initial binding of a CTC to the endothelial walls of blood vessels, and transmigration across the endothelium into the surrounding tissue. Different theories attempting to explain the metastatic formation were put forward, with tumor cells constantly maintaining a pivotal role across the hypotheses. At the first instance, the “seed and soil” theory was postulated in 1889 by Stephan Paget, who conducted postmortem research involving a cohort of 735 women with metastatic breast cancer and found that the organ distribution of metastases was not casual. In fact, based on his theory, tumor cells detaching from the primary tumor (the “seed”) grow preferentially in specific organs characterized by a suitable microenvironment (the “soil”), so that metastasis formation is not orchestrated by random events [127]. Other theories contested Paget’s proposal, supporting the anatomical/mechanical hypothesis: the arrest of cancer cells in nonspecific organs is governed by properties associated with the anatomy (i.e., venous drainage) rather than the microenvironment characteristics [128]. More recently, several studies have confirmed the validity of both the “seed and soil” and the anatomical hypotheses, suggesting that they are not mutually exclusive [128]. 

Currently, it is a matter of fact that the condition of the metastatic niche is a deciding factor in the CTCs’ fate, as they can enter a state of dormancy until the shift to a favorable microenvironment. 

### 4.2. Interaction of CTCs with Blood Cells

As reported above, CTCs face critical challenges in the bloodstream. Therefore, understanding their interactions in the liquid microenvironment could be useful to deepen the understanding of the mechanisms underlying metastatic progression [129].

The interaction of CTCs and other blood cells (i.e., immune and stromal cells) was demonstrated to be critical to escape the immune system and to permit the survival of cancer cells in circulation. These interactions can occur: (1) as a direct cell–cell interplay (CTC clusters), or (2) indirectly, through the release of specific molecules, resulting in the manipulation of normal cell functions and allowing CTC survival and extravasation in distant organs to perpetrate metastasis. In this context, the immune system plays a crucial role in tumorigenesis, as the interaction between CTCs and immune cells in circulation was demonstrated to modulate immune surveillance. Therefore, a deeper exploration of these crosstalk scenarios may be useful to better understand how CTCs respond to immunotherapy regimens [12]. 

Here, we will focus on the interactions between CTCs and platelets, macrophages and neutrophils, and on the immune-related consequences of these interplays (Figure 2).

#### 4.2.1. Platelets

Platelets are tiny anucleated cytoplasmic fragments (measuring 2–5 μm in diameter) derived from megakaryocytes, with a concentration in the bloodstream estimated to be around (150–400) × 10^9^ cells l^−1^l Although they are commonly known as mediators of blood clotting, platelets also participate in inflammation, angiogenesis, and innate immunity [130,131]. While these cells are involved in the maintenance of physiologic homeostasis, their contribution to cancer-related processes currently represents a matter of fact, as they are primary mediators of hematogenous metastasis. In patients with thrombocytosis, involving an excess of platelets in the bloodstream, prognosis is usually considered unfavorable [132]. A huge body of experimental findings has described the critical role of platelets in mediating immune evasion of cancer cells [129]. In addition, the interaction between CTCs and platelets has been investigated in the context of immune modulation and immune clearance evasion. This role has also been described at the intratumoral level: in NSCLC patients, this interaction leads to PD-L1 ingestion and presentation on the platelets’ surface, in the TME, and in the bloodstream, resulting in CD4+ and CD8+ inhibition [133]. Platelets represent one of the first circulating cell populations to interact with cancer cells, even at the intratumoral level. However, platelets have been found to interact with cancer cells in circulation by creating platelet-enriched thrombi surrounding CTCs in the bloodstream, offering physical protection from fluid shear stress, as described by Chivikula et al. [134]. At the circulation level, some studies have found that platelet cloaking on CTCs may hide CTCs from classical antibody detection [135]. Moreover, platelets can aid CTCs to evade the cytolytic activity of natural killer cells through the creation of a fibrinogen-enriched “shield” [136], and by transferring MHC-I to CTCs, preventing their identification, thus impairing natural killer cell immune surveillance [137]. 

Platelets were found to interact with CTCs and also impact their survival through the secretion of growth factors, such as TGF-β, an important regulator of immune tolerance contained in α-granules released during platelet activation. Based on the findings of Labelle and colleagues, platelets represent one of the first sources of TGF-β in circulation, which in turn activates the Smad and NF-κB signaling pathways in tumor cells, and promotes the transdifferentiation of CTCs into a mesenchymal-like phenotype [138,139,140]. Several studies have proposed the blockade of platelet–CTC interaction as an anti-metastasis treatment strategy, such as involving anticoagulants [141]. More recently, Li et al. developed genetically engineered platelets expressing the TNF-related apoptosis-inducing ligand (TRAIL) to induce CTC apoptosis, acting as a Trojan horse for CTC neutralization after their interaction [142]. 

#### 4.2.2. Macrophages

Macrophages are cells derived from myelomonocytic precursors with a critical role in physiologic processes, as they patrol for pathogens and eliminate dead cells. In the tumor microenvironment, they are one of the most representative cell populations, known as TAMs, characterized by different phenotypes and functions based on the signals that shape the TME. More specifically, TAMs can polarize toward M1-like and M2-like macrophages, with inhibitory and growth-promoting properties, respectively [143]. Both tumor-resident and circulating TAMs have a huge impact on cancer cells, especially through the release of a plethora of factors (polypeptides, metabolites, cytokines, and others) with tumor-promoting and tumor-protecting properties, and by fusing with tumor cells [144]. Their role has been found to be involved not only with primary tumor cells, but also with CTCs. For instance, in colorectal cancer, the presence of CD163+ TAM in the tumor invasive front has been associated with the presence of epithelial-to-mesenchymal transition (EMT) in CTCs, leading to increased motility and metastatic power [145]. In this context, Cheng Wei and collaborators described a positive feedback loop between cancer cells and TAMs involving the IL6/STAT3 pathway, with a critical role in cancer progression and metastasis [146]. It has been proposed that CTCs communicate directly with macrophages via the CD47/SIRPα axis. CD47 is a “don’t-eat-me” signal and its overexpression results in the repulsion of phagocytic attacks, thus evading the immune response. CD47 binds to its receptor signal regulatory protein α (SIRPα), which is expressed by macrophages and dendritic cells, conferring CTCs with a non-immunogenic profile [147,148]. 

The interaction of CTCs with macrophages was shown to be involved in several processes, including migration, invasion, and immunosuppression, to escape from antitumor immune responses. For instance, by co-culturing peripheral blood mononuclear cells with CTC cell lines derived from small-cell lung cancer patients, Hamilton and co-workers investigated the interplay between macrophages and CTCs. They found that CTC-derived factors stimulate the differentiation of monocytes toward CD14+, CD163^weak^, and CD68+ TAM of the M2-like type, resulting in immunosuppression. In addition, migration and invasion are enhanced by CTCs that elicit the secretion of factors that drive migration and invasion, such as osteopontin (OPN), monocyte chemoattractant protein 1 (MCP-1), IL-8, chitinase3-like 1 (CHI3L1), platelet factor (Pf4), IL-1ra, matrix metalloproteinase 9 (MMP-9), and others [149]. Moreover, it has been demonstrated that cancer patients can display, in circulation, hybrid cells deriving from the interaction between macrophages and cancer cells, including tumor resident cells, with a critical role in therapy response and immune evasion [150]. These cells can derive from the phagocytosis of cancer cells (cancer-associated macrophage-like (CAML) cells) and from macrophage–cancer cell fusion (circulating hybrid cells, CHCs). CAML cells are immune cells that do not recapitulate tumorigenesis and originate from the phagocytosis of cancer cells. They have been found in the circulation of patients with different tumor types, and an increase in CAML cell numbers has been observed in patients responding to chemotherapy [151]. The number of CAML cells was shown to be inversely correlated with the number of CHCs, which in turn is associated with decreased therapy response. In fact, CHCs are giant hybrid cells (size ≥ 30 μm) with both epithelial and myeloid phenotypes (dual CK+/EpCAM+ and CD14/CD45+), characterized by pro-metastatic power, chemotherapy resistance, and immune tolerance [150,152,153].

#### 4.2.3. Neutrophils

Neutrophilic cells constitute the most represented myeloid population in human blood. They are rapidly recruited to sites of tissue injury and, in the case of microbial infection, by signals that include hydrogen peroxide, chemokines, and cytokines [154]. In recent years, a growing body of evidence has leveraged their role as regulators of cancer and, by virtue of their presence in circulation, the interaction between neutrophils and CTCs has been investigated as well [155]. According to the literature, the interplay between neutrophils and CTCs can occur in two manners: (1) by direct interaction, forming clusters, and (2) through the creation of web-like structures, called neutrophil extracellular traps (NETs), formed by DNA–histone complexes and activated neutrophil-derived proteins [156].

It has been shown that the interaction of CTCs with neutrophils starts at the early stages of tumor cell migration in the primary tumor, therefore tumor-resident neutrophils detach together with cancer cells and enter the bloodstream in the form of clusters. Neutrophils interacting with CTCs support cell cycle progression in circulation and enhance the metastatic power. Moreover, this interplay is VCAM-1-dependent, presumably by binding to the integrins expressed by neutrophils, and its inhibition prevents the formation of clusters [157]. Gene expression analyses performed on breast cancer patients showed that CTC-associated neutrophils might be polarized towards the N2-like phenotype [158]. In fact, neutrophils are characterized by high plasticity and can be polarized into antitumor N1-like cells, as well as pro-tumor N2-like cells, depending on the environmental factors [154]. Compared to N1-like neutrophils, N2-polarized neutrophils do not produce high levels of proinflammatory agents, such as chemokines (such as CCL3, CXCL9) or cytokines (IL-12, TNF-α, and others), while they are responsible for the secretion of high levels of arginase that can inactivate T cell functions [159]. 

On the other hand, neutrophils have also been described in association with CTCs in the form of NETs, which are produced when neutrophils are exposed to certain stimuli in response to bacterial infections [160]. NETosis consists of the expulsion of NETs out of neutrophils to the extracellular space, as the result of surgical trauma and injuries. Due to its pro-tumor activity, NETosis plays a fundamental role in cancer progression and metastasis, as intensively reviewed by Kwak et al. [161]. It has been shown that NETs, together with platelets, can exploit their sticky jelly-like properties to wrap up CTCs, acting as a shield, protecting them from FSS and immune attacks [162].

### 4.3. CTCs’ Role in Immunotherapy Response Prediction

The need for biomarkers capable of assessing the treatment response and predicting which patients would benefit from immunotherapy is unquestionable. Several studies have attempted to settle the role of CTCs as an accessible and minimally invasive surrogate of the tumor for biomarker investigation, especially when a tumor biopsy is not performable or the patient is unfit [163]. In addition, CTC analysis also provides a snapshot of biomarker expression in longitudinal studies, allowing the response to therapy to be monitored [164,165].

#### 4.3.1. EpCAM-Based CTC Enumeration

Since the early 2000s, CTC enumeration has been demonstrated as a valuable approach for prognosis determination and therapy monitoring in cancer patients. More specifically, in 2004, Cristofanilli et al. reported for the first time, in a prospective study involving 177 metastatic breast cancer patients, that the number of CTCs assessed using the CellSearch system is an independent prognostic factor, with patients having equal or greater than 5 CTC/5.5 mL of blood displaying a lower PFS [166]. For now, CellSearch remains the only FDA-cleared CTC-based assay for the determination of prognosis in patients with advanced epithelial tumors. This platform relies on ferrofluid-based capture reagents targeting the EpCAM antigen for capturing CTCs. In the following years, CTC enumeration has been investigated as a potential tool to monitor the therapy response in several trials, including in regard to immunotherapy as well. 

In a study by Tamminga et al., CTCs were enumerated using CellSearch in 104 advanced NSCLC patients, receiving checkpoint inhibitors at the baseline (T0) and at 4 weeks of treatment (T1). The durable response rates were twice as high in CTC negative patients at T0, and six times as high in patients that in T1 had decreased CTCs compared to patients with stable or an increased number of CTCs. No predictive role was found related to the early tumor response [167]. Similar findings in advanced NSCLC patients receiving PD-1/PD-L1 inhibitors were reported by Park and colleagues. The enumeration of CTCs using a size-based system (CD-PRIME™, Clinomics Inc., Ulsan, Republic of Korea) revealed that a high count of CTCs during treatment (not the baseline) and an increased number of CTCs predicted disease progression. This trend was also observed with an increasing number of noncanonical EpCAM- and CD45-positive CTCs (double positive CTCs) [168], whose presence has been described in different tumor types, but their role is not yet fully elucidated [169]. 

#### 4.3.2. PD-L1 Determination

PD-L1 (programmed cell death ligand 1) acts as a critical regulator of immune tolerance through the interaction with PD-1 and it is one of the most studied markers in patients treated with immunotherapy. The function of this checkpoint acts as a “don’t-find-me” signal to the adaptive immune response [170]. PD-L1 has been established as a predictive biomarker of the response to PD-L1/PD-1 inhibitors [171]. However, it is reported that >50% patients with high PD-L1 expression in tumor tissue do not benefit from first-line Pembrolizumab [172], and 10% of patients with a PD-L1 negative tumor do respond to second-line inhibitors [173]. The reason could be attributed to the spatiotemporal heterogeneity of PD-L1 in tumor tissue [174]. Studies in the last few years have attempted to investigate the role of CTC PD-L1 determination in predicting the therapy response, but with contrasting findings. The different approaches for CTC enrichment and PD-L1 detection, the different cancer types, and therapies used, may be the basis of the inconsistent data. 

A proof-of-concept for PD-L1 determination in CTCs for the stratification of patients receiving immunotherapy was provided by Ilié et al. More specifically, they found that in advanced NCSLC patients the concordance of PD-L1 expression in CTCs and matched tumor tissue was 93%, supporting the potential of CTCs as a real-time “liquid biopsy” [170]. In their study, CTCs were enriched from peripheral blood based on their size and deformability using the ISET filtration platform, and PD-L1 immunostaining was directly performed on filters [170]. 

Other studies were conducted on different tumor types to look into the predictive role of PD-L1 expression in CTCs. Being historically involved in immunotherapy [20], several studies have focused on PD-L1 determination in CTCs from lung cancer. 

In a retrospective study on advanced NSCLC patients by Guibert et al., the expression of PD-L1 was assessed in CTCs isolated before Nivolumab treatment and during progression. They found that CTCs had higher PD-L1 positivity rates compared to matched tissues (83% vs. 41%), but, conversely to the abovementioned study, higher baseline PD-L1+ CTCs levels were observed in non-responder patients, meaning those that experienced PFS < 6 months [174]. The investigation by Sinoquet et al. [175], using CellSearch for 54 patients with advanced NSCLC, revealed instead a low concordance of PD-L1 expression in CTCs and matched tumor tissues (54%). Nicolazzo et al. evaluated the expression of PD-L1 in CTCs from NSCLC patients at the baseline, and at 3 and 6 months after starting Nivolumab treatment, using the EpCAM-based CellSearch enumeration platform. They found that at the baseline and at 3 months, almost all the patients had PD-L1 + CTCs (100% prevalence). In addition, patients displaying PD-L1 negative CTCs at 6 months experienced a clinical benefit, while those patients with PD-L1+ CTCs experienced disease progression, suggesting that this phenomenon could act as a mechanism of therapy escape and dichotomize the patients [176]. Based on their findings, the assessment of PD-L1 expression on CTCs acquires a more predictive value late in course of treatment (at 6 months) due to the biological time required by the immune system to eliminate PD-L1+ CTCs, while PD-L1 determination at 3 months is less informative [176]. In another study on advanced NSCLC patients treated with PD-L1/PD-1 inhibitors as the second- or third-line treatment, Dall’Olio et al. found that PD-L1+ CTCs were associated with better survival and worse survival when measured during pre-treatment and post-treatment, respectively [177]. In this study, the CellSearch platform was used. In a study conducted on 47 hepatocellular cancer (HCC) patients receiving PD-1 inhibitors in combination with radio- and antiangiogenetic therapy, the identification of <2 PD-L1 positive CTCs using CytoSorter at the baseline was associated with a higher objective response rate. In addition, patients with a dynamic decrease in PD-L1 positive CTCs at 1 month after treatment were more likely to display an objective response [178]. In a study by Tan and colleagues, the expression of PD-L1 in CTCs from 155 patients with different advanced tumor types was assessed to investigate its role in predicting and monitoring the response to PD-1/PD-L1 blockade immunotherapies. The disease control rate (DCR) was significantly different in patients with PD-L1+ CTCs (39.29%) and PD-L1- CTCs (71.65%). Moreover, a higher number and percentage of PD-L1+ and PD-L1-high CTCs were observed at the baseline in patients in the partial response and stable disease group, compared to those with progressive disease. In hepatocellular carcinoma patients, a higher baseline count of PD-L1-high CTCs was associated with a clinical benefit, suggesting the potential use of PD-L1 determination on CTCs to predict the response to anti-PD-1 and anti-PD-L1 monoclonal antibodies [172]. In a study conducted on 40 patients with a melanoma diagnosis, Khattack et al. found that the number of PD-L1+ CTCs was higher in patients responding to Pembrolizumab compared to non-responders at the baseline [179]. 

However, some studies have shown that determination of PD-L1 expression in CTCs is not useful to predict treatment outcome. In the phase 2 INTEGA trial conducted on esophagogastric patients, PD-L1 expression on CTCs using CellSearch was not associated with a specific outcome in the Ipi arm (Trastuzumab and Nivolumab in combination with Ipilimumab) [171]. In metastatic renal cell carcinoma patients, Bootsma et al. focused on the expression of PD-L1 and HLA-I in CTCs, captured and enriched using a versatile exclusion-based rare sample analysis (VERSA). In fact, given their diametrical effects on immune evasion, PD-L1 and HLA-I expression levels were contextualized by computing a HP ratio. In patients receiving ICB, the HP ratio was found to decrease due to the clearance of tumor cells characterized by high HLA-I and low PD-L1 levels, while patients with an increasing HP ratio over time had worse outcomes. Therefore, this approach could provide a predictive role of the HP ratio in the prediction and monitoring of mRCC patients [180].

## 5. Future Applications of EVs and CTCs in Immunotherapy

The power of EVs and CTCs in liquid biopsies has currently achieved stunning outputs, but their introduction in clinical practice is still far from a reality due to several reasons, including the high costs, a lack of standardization, and suboptimal specificity and sensitivity. Nevertheless, in recent years researchers have attempted to encourage the development of new applications involving CTCs and EVs.

In this context, the engineering of EVs has recently been described as one of the most promising tools in immunotherapy, thanks to their ability to modulate the TME, and the stability in body fluids, including peripheral blood [181]. Moreover, EVs have a natural aptitude for crossing cell membranes and barriers, overcoming the issue related to the therapeutic delivery of bioactive molecules (DNA, RNA, and proteins) to some body compartments, such as the central nervous system [182]. In immunotherapy, the application of engineered EVs can be useful as an immunomodulatory strategy to improve treatment efficiency and to rewire antitumor immunity [182]. In 2021, engineered EVs overexpressing a high-affinity variant human PD-1 protein (havPD-1) were shown to have efficacy as a direct immunotherapy agent and in combination with other agents (i.e., PARP inhibitors). In xenograft tumor models, havPD-1 EVs were shown to induce apoptosis of PD-L1 overexpressing tumor cells and activating cytotoxic T cells in xenograft models [183,184]. Moreover, the encapsulation of chemotherapeutic agents within engineered EVs showed an increased antitumor effect compared to monotherapy, due to the immunostimulatory function of chemotherapeutic agents [184,185]. Engineered EVs have also been demonstrated as a promising strategy to repolarize TAMs, enhancing T cell antitumor immunity and ICI efficacy [186]. In this context, Choo et al. demonstrated that engineered EVs derived from M1 macrophages are able to reset M2-like TAMs (pro-tumorigenic properties) to the M1 phenotype (tumor inhibition properties) in vitro and in vivo, potentiating the antitumor efficacy of ICI treatment [187]. However, although most scientific evidence supports the role that cancer-derived EVs have in suppressing the antitumor response, it is known that they can also have an immunostimulatory role that can boost the efficacy of immunotherapy for patients undergoing treatment [188]. Nowadays, there are some main lines of action for immunotherapy that involve cancer-derived EVs, such as the suppression approach of cancer-derived EVs’ secretion, then also the enhancing of the immunostimulatory factors around the EVs’ surface and, lastly, the use of cancer-derived EVs as carriers in the field of vaccines. The potential of using these cancer-derived EVs is still far from clinical use, even with its huge potential [188]. 

On the other hand, while EVs are still far from being introduced in clinical practice, CTC enumeration using CellSearch is used for prognosis assessment for some advanced tumors. However, since this platform allows for the detection of exclusively EpCAM+ cells, it has been reported that CellSearch may fail to detect CTCs with unconventional characteristics, such as low/absent EpCAM expression, or the expression of mesenchymal markers due to EMT (non-conventional CTCs, ncCTCs) [189,190]. The interaction between CTCs with other blood cells, such as platelets, can induce EMT-like features in CTCs [191]. EMT-like CTCs have been described in regard to different tumor types in association with an invasive phenotype [192,193]. Thanks to the extraordinary advances achieved in the fields of transcriptomics and genomics, it is now possible to better understand the molecular mechanisms orchestrating tumor immune escape, as well as the interplay between CTCs and immune cells. In future, high-throughput and computational technologies could allow the discovery of novel markers and potentially immunogenic antigens on CTCs with a predictive and/or actionable role in immunotherapy [194]. 

## 6. Conclusions

Both EVs and CTCs have direct and indirect interactions with the immune system. By analyzing the main studies addressing the relationships between the immune system and EVs and CTCs we could conclude that this is a new frontier in cancer research, with a promising impact in the clinic. CTCs and EVs could represent biomarkers both capable of identifying patients responsive or resistant to immunotherapy, and targets for therapeutic approaches. However, for a possible transfer from the laboratory bench to the patient’s bedside, other studies should be conducted to clarify and confirm their potential and to better define the clinical contexts in which they could be developed. Despite numerous attempts to study the mechanism by which both EVs and CTCs influence immunotherapy in depth, several mechanisms have still not been deeply examined. Future studies will be decisive in providing answers to these unresolved questions, offering useful evidence to determine the predictive role of EVs and CTCs.

## Figures and Tables

**Figure 1 cells-13-00337-f001:**
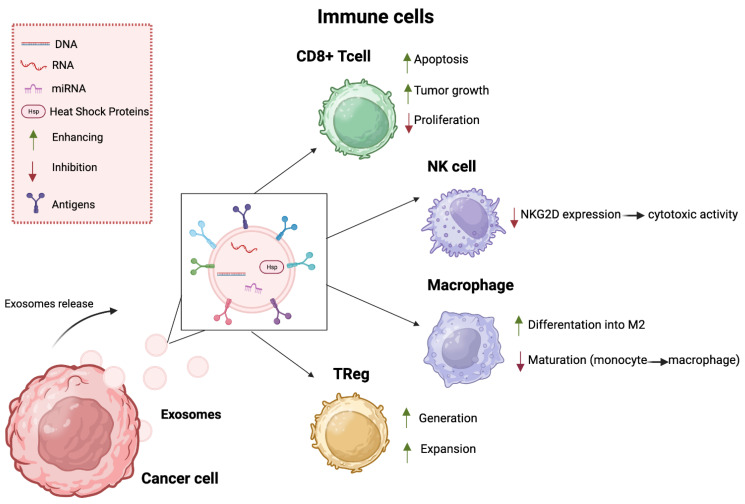
Representation of the influence that exosomes have on the cells in the immune system. In some cases, they inhibit the activity of the cells that try to hinder the tumor; in others, they promote cancer by favoring those cells that encourage tumor growth. Treg: regulatory T cell; Nk: natural killer; miRNA: microRNA. Created using BioRender.

**Figure 2 cells-13-00337-f002:**
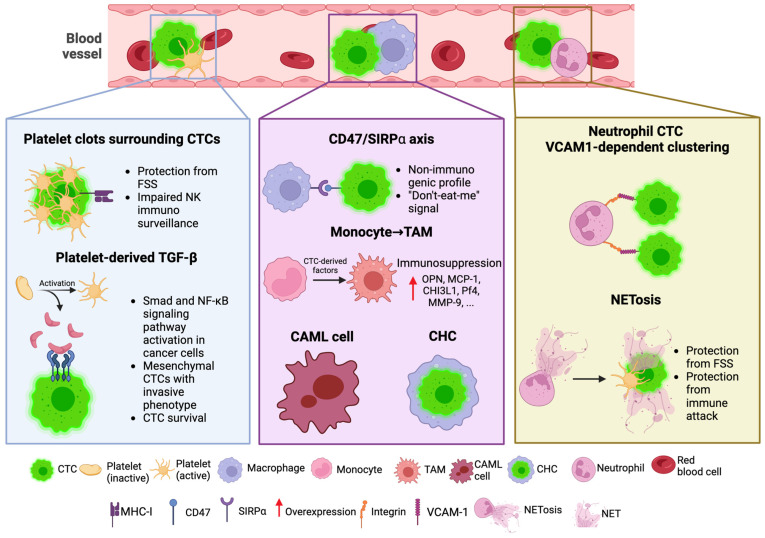
The interactions between CTCs and other blood cells: platelets, macrophages, and neutrophils. CTC: circulating tumor cells; TGF-β: transforming growth factor-β; FSS: fluid shear stress; NK: natural killer; TAM: tumor-associated macrophage; CAML: cancer-associated macrophage-like cells; CHC: circulating hybrid cell; NET: neutrophil extracellular trap. Created using BioRender.

**Table 1 cells-13-00337-t001:** Cargo present in exosomes, with an implication for immunotherapy in different cancer types.

Exosome Cargo	Activity in Tumors	Tumor Types	Reference
**Exosomal Proteins**			
PD-L1	Suppress CD8+ T cells, it is correlated with an anti-PD-1 response	Melanoma	[84]
	Related to the disease activity and to the clinical stages	H and N cancer	[103]
	PD-L1 levels were effective for predicting anti-PD-1 therapies	NSCLC	[104]
	Correlated with the worse survival rate	PDAC	[105]
**Exosomal lipids**			
Phosphatidylcholine	Resistance to immunotherapy	B cell lymphoma	[106]
**miRNAs**			
hsa-miR-320 (hsa-miR-320d, hsa-miR-320c, hsa-miR-320b)	Potential predictors for immunotherapy response	NSCLC	[107]
hsa-miR-125b-5p	Potential target for anti-PD-1 treatment(its decrease is the mark of a better outcome and a longer PFS)		[107]
hsa-miR-34a	Associated with the response to immunotherapy and the outcome		[108]
**Circular RNAs**			
circCCAR1	Promotes CD8 + T cell dysfunction and anti-PD1 resistance	HCC	[109]
circUHRF1	Induces natural killer cell exhaustion and resistance to anti-PD-1 therapy		[110]
circUSP7	Induces CD8+ T cell dysfunction and anti-PD1 resistance	NSCLC	[111]

PD-L1: programmed death-ligand 1; PD-1: programmed death 1; PDAC: pancreatic ductal adenocarcinoma; HCC: hepatocellular carcinoma; H and N: head and neck; NSCLC: non-small-cell lung cancer; miRNA: microRNA.

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
