# Peer review of "Extracellular Vesicles, Circulating Tumor Cells, and Immune Checkpoint Inhibitors: Hints and Promises"

_cells, 2024, doi:10.3390/cells13040337_

Round 1

Reviewer 1 Report

Comments and Suggestions for Authors

Dear authors,

thank you for this outstanding and comprehensive review. I have a few minor comments. 

1. The description of immune checkpoint inhibitors takes about 10% of all text, however, it is stated in the title. I would suggest to add some details about the mechanism of action of ICIs and corresponding specific treatment. In comparison with two other parts (EVs and CTCs) the part of ICIs seems to be quite laconic at the moment.

Other comments are rather technical.

2. Please, correct the reference type on lines 88-91.

3. Please, check the abbreviation you've used. Sometimes you are introducing them more than once. E.g. TDSFs (lines 279, 287), TAMs, BMDCs and others.

4. What IL do you mention on line 289?

Author Response

Replies to Reviewer 1
Dear authors,
thank you for this outstanding and comprehensive review. I have a few minor comments.
1.The description of immune checkpoint inhibitors takes about 10% of all text, however, it is stated in the title. I would suggest to add some details about the mechanism of action of ICIs and corresponding specific treatment. In comparison with two other parts (EVs and CTCs) the part of ICIs seems to be quite laconic at the moment.
Reply: we thank the reviewer for the precious suggestion. We modified the text accordingly by adding some details of ICI mechanisms of action (lines 114-126).
Other comments are rather technical.
2. Please, correct the reference type on lines 88-91.
3. Please, check the abbreviation you’ve used. Sometimes you are introducing them more than once. E.g. TDSFs (lines 279, 287), TAMs, BMDCs and others.
4. What IL do you mention on line 289?
Reply: we apologize for the missing information. We modified the text accordingly.

Reviewer 2 Report

Comments and Suggestions for Authors

In this manuscript, Bandini and colleagues discuss the recent advances in the identification of immune checkpoints using extracellular vesicles (EVs) and circulating tumor cells (CTCs) with their potential applications as predictive biomarkers for immunotherapy, and their prospective use as innovative clinical tools. 

The review is well written, and supported by pertinent literature. The provided references are appropriated and up to date. 

Some comments: 

-mention of MISEV guidelines, fundamental for EVs, studies are missing  

-significantly important part regarding the role of cancer-derived EVs following anti-cancer treatment as booster for immunotheray is missing

Author Response

Replies to Reviewer 2
In this manuscript, Bandini and colleagues discuss the recent advances in the identification of immune checkpoints using extracellular vesicles (EVs) and circulating tumor cells (CTCs) with their potential applications as predictive biomarkers for immunotherapy, and their prospective use as innovative clinical tools.
The review is well written, and supported by pertinent literature. The provided references are appropriated and up to date.
Some comments:
-mention of MISEV guidelines, fundamental for EVs, studies are missing
Reply: we thank the reviewer for the comments. We mentioned the MISEV guidelines in accordance with the suggestion in lines 485-489.
-significantly important part regarding the role of cancer-derived EVs following anti-cancer treatment as booster for immunotheray is missing

Reply: we thank the reviewer for the suggestion. We discuss this point in the section dedicated to future applications (lines 1269-1277).

Reviewer 3 Report

Comments and Suggestions for Authors

In the MS by Bandini et al., the authors made a data literature review on the role of extracellular veisicles (EVs) as biomarkers and/or biotools in the antitumoral therapy using immunological checkpoint inhibitors (ICIs). Authors illustrated the differen ICIs and the use of anti-ICIs antibodies in the antitumoral therapy then they made an overview of the different types of EVs and their role in the settlement of metastatic niche as well as their interplay with the different immune cells and their cargo. Finally they illustrated the Circulating Tumor Cells (CTCs), how they interact with the immune cells and how they could be exploited, together with EVs as diagnostic tools. A small paragraph is dedicated to the possible application of modified EVs as promising tools for cancer therapy.

The review made by Bandini et al., is well conceived and organized and the authors developed the topic in an extensive way and in a reasoned manner so in my opinion it is already suitable for publication  in Cells. Just few typos need to be revised:

Line 338: change miRNa with miRNA

Line 392: add space after the ref (92)

Line 569 : change “(150-400)x109 cells l-12 with “(150-400)x109 ml-1

Line 719: change IT with it

Lines 730-732: check and eventually revise the sentence “More specifically….. (165)”

Author Response

Replies to Reviewer 3
In the MS by Bandini et al., the authors made a data literature review on the role of extracellular veisicles (EVs) as biomarkers and/or biotools in the antitumoral therapy using immunological checkpoint inhibitors (ICIs). Authors illustrated the differen ICIs and the use of anti-ICIs antibodies in the antitumoral therapy then they made an overview of the different types of EVs and their role in the settlement of metastatic niche as well as their interplay with the different immune cells and their cargo. Finally they illustrated the Circulating Tumor Cells (CTCs), how they interact with the immune cells and how they could be exploited, together with EVs as diagnostic tools. A small paragraph is dedicated to the possible application of modified EVs as promising tools for cancer therapy.
The review made by Bandini et al., is well conceived and organized and the authors developed the topic in an extensive way and in a reasoned manner so in my opinion it is already suitable for publication in Cells. Just few typos need to be revised:
Line 338: change miRNa with miRNA
Line 392: add space after the ref (92)
Line 569 : change “(150-400)x109 cells l-12 with “(150-400)x109 ml-1
Line 719: change IT with it
Lines 730-732: check and eventually revise the sentence “More specifically….. (165)”
Reply: we thank the reviewer for the comments and we apologize for the typos. We corrected the text accordingly.